# Aspects of Parent–Child Interaction from Infancy to Late Adolescence are Associated with Severity of Childhood Maltreatment through Age 18

**DOI:** 10.3390/ijerph17113749

**Published:** 2020-05-26

**Authors:** Jennifer E. Khoury, Mallika Rajamani, Jean-François Bureau, M. Ann Easterbrooks, Karlen Lyons-Ruth

**Affiliations:** 1Department of Psychiatry, Cambridge Hospital, 1035 Cambridge Street, Cambridge, MA 02141, USA; mrajamani26@gmail.com (M.R.); klruth@hms.harvard.edu (K.L.-R.); 2Department of Psychiatry, Harvard Medical School, Boston, MA 02115, USA; 3School of Psychology, University of Ottawa, Ottawa, ON K1N 6N5, Canada; Jean-Francois.Bureau@uOttawa.ca; 4Eliot-Pearson Department of Child Development, Tufts University, Medford, MA 02155, USA; ann.easterbrooks@tufts.edu

**Keywords:** child maltreatment, parent–child interaction, attachment, longitudinal

## Abstract

Childhood maltreatment (CM) is a pervasive public health problem worldwide, with negative health consequences across the lifespan. Despite these adverse outcomes, identifying children who are being maltreated remains a challenge. Thus, there is a need to identify reliably observable features of parent–child interaction that indicate risk for CM and that can instigate strategically targeted family supports. The aim of this longitudinal study was to assess multiple aspects of observed mother–child interaction from infancy to late adolescence as risk indicators of the overall severity of CM by age 18. Mother–child dyads were assessed in infancy (*N* = 56), at age 7 years (*N* = 56), and at age 19 years (*N* = 56/110). Severity of CM through age 18 was indexed by combined prospective and retrospective assessments. Interactions associated with severity of CM by age 18 included maternal hostility in infancy, maternal withdrawal in infancy and middle childhood, child disorganized attachment behavior in middle childhood and late adolescence, as well as hostile and role-confused interactions in late adolescence. This study identifies new indices of maternal and child behavior as important risk indicators for the severity of CM. These indices could be used to improve early identification and tailor preventive interventions for families at risk for CM.

## 1. Introduction

Childhood maltreatment (CM), including emotional and physical abuse or neglect, witnessed domestic violence, and sexual abuse, has significant negative implications for child development [1]. Despite elevated awareness of CM as a global problem, prevalence of CM continues to be high, ranging from 12–36% worldwide [2]. In addition, children continue to endure long-lasting psychological and social consequences of maltreatment [3]. In the United States in 2018, 678,000 children were identified as victims of abuse or neglect and 1770 children died from abuse or neglect [4]. CM is also a serious global public health problem because maltreatment predicts later aggression, antisocial behavior, and delinquency [5]; high-risk sexual behaviors [6]; smoking [7]; obesity [8]; substance use problems [9,10]; and a range of mental health problems, including suicide [11,12]. Maltreatment also increases risk for a range of physical health problems, including cardiovascular disease and cancer [13,14].

Despite the global health implications of CM, child abuse and neglect can be hidden from observation. Thus, rates of CM documented by Child Protective Services (CPS) in childhood are consistently much lower than rates of maltreatment reported retrospectively [4,15]. Because of the difficulties in identifying maltreatment in childhood [15], most research assessments of CM are self-report instruments administered in adulthood. While such instruments are likely to remain first-line measures for assessing prevalence and correlates of maltreatment, they have been criticized for being open to biased recall [16,17].

Given the pervasive consequences of CM and the intractable issues associated with early identification of maltreatment, it is also important to study observable prospective indicators in childhood that are associated with cumulative severity of CM. The identification of observable forms of child and parent interaction that are indicative of severity of CM can provide a set of risk indicators that can be reliably assessed across childhood, and that can qualify families for support and intervention early in the child’s development, whether or not CM itself is documented by social service workers.

The aim of the present study was to assess multiple aspects of observed maternal and child interaction from infancy to late adolescence in order to identify features of interaction that might serve as prospective or concurrent indicators of the overall severity of CM by age 18. Of particular interest was whether there are consistent aspects of parent–child interaction across development that are associated with overall severity of CM by age 18.

### 1.1. Attachment Theory: A Guiding Framework

Given that the parent–child relationship provides the underlying context for healthy development, an attachment framework is useful in understanding how disrupted parent–child interactions might be related to CM. According to attachment theory [18], the attachment system evolved in order to increase survival, such that infants seek proximity to their caregivers to attain care and protection during times of distress and threat. The early parent–child relationship shapes the ways in which children form mental representations (internal working models) about themselves, their caregivers, and other meaningful relationships throughout life.

Forming a secure attachment relationship with one or more caregivers is a primary developmental milestone of infancy. Ainsworth and colleagues (1978) established reliable observational methods for detecting individual differences in attachment security and organization in infancy [19]. In prior meta-analyses, children who were being maltreated, as assessed by protective service involvement, were found to exhibit more insecure and disorganized forms of attachment behavior with their mothers [20,21]. In other studies, children classified as having organized patterns of attachment behavior (secure, avoidant, and ambivalent) have been shown to demonstrate less severe psychological difficulties compared to children with disorganized attachments (see meta-analyses [22,23,24]). In contrast, children with disorganized attachment patterns do not display a coherent strategy when seeking comfort from their caregivers. Instead, they display a variety of conflict behaviors or other odd, out-of-context behaviors in the presence of the parent when stressed. These behaviors may include indicators of fear, odd behaviors such as freezing all movement, or unpredictable alternations in approach and avoidance behavior in the presence of the caregiver [25].

Importantly, meta-analyses have confirmed that caregiver behavior predicts infant attachment patterns, such that caregivers who are sensitive to a child’s needs and consistently responsive to the child’s cues are more likely to have infants who develop a secure attachment pattern, while less sensitive caregiving behavior has been linked to insecure but organized attachment patterns (avoidant and ambivalent) [26]. In contrast, more disrupted forms of parenting behavior, including hostility, disorientation, and withdrawal in interaction with the infant, have been associated with disorganized infant attachment strategies (meta-analysis, [27]). Given the large number of studies connecting concurrent, protective service-identified CM with disorganized attachment behavior in infancy (meta-analyses, [20,21]), attachment disorganization and associated disrupted caregiving behavior are important candidates for observable variables across childhood and adolescence with demonstrated implications for child risk. We examine these behaviors in the current investigation and discuss them in more detail below.

While early research focused primarily on attachment-related interactions in infancy and preschool, more recent work has extended attachment assessments to middle childhood and adolescence. This work has revealed that during the preschool years, some children who were previously disorganized now actively attempt to control the parent’s attention and behavior through punitive or caregiving behavior, while others continue to display disorganization (odd, out-of-context, or disoriented behaviors in interaction) [28]. These controlling-punitive or controlling-caregiving stances have been further documented in both middle childhood [29] and in adolescence [30]. At these later ages, both controlling behaviors and continued signs of disorganization have been associated with maladaptive outcomes. Studies of preschool and early school age children report associations between controlling and disorganized attachment behaviors and externalizing and/or internalizing behavior problems [31]. In early adolescence, disoriented dyads at age 13 showed higher levels of teacher-rated internalizing problems at age 15, while male adolescents in role-confused dyads at age 13 reported higher levels of involvement in risk behaviors by age 15, including unprotected sexual activity and substance use problems [32]. In late adolescence, disorientation (odd, out-of-context behavior) at age 19 was predicted by attachment disorientation in infancy, and both disorientation and controlling forms of behavior were associated with poorer quality intimate partner relationships and increased depressive and dissociative symptoms [30].

### 1.2. Attachment Patterns and Child Maltreatment

Given the importance of the quality of the early mother–child interaction in fostering a secure attachment relationship, it is not surprising that children who experience CM are at elevated risk for showing insecure and disorganized attachment behavior. For example, a recent meta-analysis found that maltreated children showed fewer secure (*d* = 2.10) and more disorganized (*d* = 2.19) attachments compared to other high-risk children (*d* = 0.48 and *d* = 0.48, respectively) who were not exposed to maltreatment [20]. This literature suggests that early signs of disorganized attachment might indicate future risk for CM. However, all of the studies included in these prior meta-analyses assess attachment patterns of concurrently maltreated children prior to four years of age. Thus, it remains unclear how predictive early parent–child interactions might be of overall severity of CM assessed at the end of adolescence, and whether attachment assessments after age four have similar relations to CM.

Despite the strong empirical support of the link between infant/preschool attachment and CM, only a handful of studies have assessed observable attachment behavior beyond the preschool period in relation to CM. Using the current study sample, Byun et al. found that severity of abuse was related to forms of dyadic attachment behavior, including lower levels of collaboration and higher levels of dyadic disorientation and hostile-punitive interaction [33]. However, it remains unclear the extent to which the adolescent’s behavior or the parent’s behavior or both are contributing to the relation with maltreatment. In related research, Zvara et al. showed that women who were sexually abused during childhood displayed more boundary dissolution (i.e., role confusion) when interacting with their offspring [34]. Taken together, this research suggests that CM is likely associated with disorganized and controlling attachment interactions from infancy through late adolescence.

### 1.3. Disrupted Parental Behavior and Child Maltreatment

In addition to infant/child attachment patterns, a large body of research examines aspects of parenting behavior as risk factors for CM [35,36,37]. In individual studies, specific aspects of observed parenting behavior have been associated with CM. For example, physically abusive mothers used more harsh discipline strategies and aggressive communication compared to non-abusive mothers [38]. In another study, mothers who maltreated their children were found to be less involved, to display fewer physical and verbal strategies to direct their child’s attention, and to display more negative behavior while interacting with their children [39]. In a meta-analysis where all cases of CM were documented by CPS, Wilson et al. found that maltreating parents were more likely to display aversive parenting behaviors (e.g., negative verbal/physical behavior and physical aggression), less likely to display positive parenting behaviors (e.g., praise, positive affect, and positive physical touch) and less likely to be involved (e.g., initiate interaction and responsiveness) with their children [40]. Further, physically abusive parents were more likely to display aversive behaviors, whereas neglectful parents were more likely to display lack of involvement [40].

Importantly, few studies assess observed parent–child interaction from ages 10 to 18 in relation to CM (meta-analysis, [40]). Furthermore, maladaptive parent and child behavior has rarely been observed longitudinally from infancy to late adolescence to examine whether there are consistent associations over time between particular aspects of parenting and CM or whether the nature of these associations vary at different points in the child’s development, as might be expected given dramatic changes in the child’s ways of engaging with others from infancy to late adolescence. Underscoring this need, Cicchetti and Doyle (2016) recently called for multi-wave longitudinal research to examine the interrelations among CM, attachment organization, and parent–child interactions [41].

### 1.4. Aims of the Current Study

To address these gaps in the literature, the purpose of the current study was to observe parent-child interactions in infancy, middle childhood, and late adolescence in order to assess whether prospective or concurrent risk indicators of overall severity of CM by age 18 could be identified over development. Severity of CM by age 18 was indexed by a multi-method index that included both prospective and retrospective assessments. The first hypothesis of the study was that child disorganized attachment behavior, assessed in infancy, middle childhood, and late adolescence, would be associated with greater severity of CM by age 18. The second hypothesis was that disrupted parenting behaviors assessed in infancy, middle childhood, and late adolescence would also be associated with greater severity of CM by age 18. Given inconsistencies in the literature regarding the relation between CM and decreases in positive maternal behaviors (e.g., sensitivity, warmth, and engagement), no hypotheses were advanced regarding whether these aspects of maternal behavior would be related to severity of CM.

## 2. Methods

### 2.1. Participants

Participants were 110 low to moderate income mother–child dyads (offspring: 59.6% females; M age = 19.9 years old; SD = 1.46). All 110 dyads were assessed in late adolescence. In addition, 56 of the 110 dyads had been assessed longitudinally in infancy and middle childhood. Associations between parent and child behavior and CM were assessed in the longitudinal sample (*N* = 56) in infancy and middle childhood and in the larger sample (*N* = 110) in late adolescence.

Among the 56 dyads seen longitudinally from infancy, family income for 56% of the sample was equal to or less than $30,000. A total of 32.1% of mothers were single parents during the child’s infancy, 61.8% were single in middle childhood, and 60% were single at the time of the late-adolescence visit.

The 54 families seen only in late adolescence were matched to the longitudinal families based on adolescent age, ethnicity, and that the mothers were single parents. However, family income was slightly higher among the cross-sectional families than the longitudinal families (family income mean range: longitudinal, $20,000–$30,000/year; cross-sectional, $30,000–$40,000/year; *F* (1, 118) = 9.63, h = 0.28, *p* < 0.01).

The 56 longitudinally studied families were part of a cohort of 76 low-income families recruited during the first 18 months of the child’s life. Half of the families seen in infancy were referred to the study by health or social service providers because of concerns about the quality of care provided to the infant. This resulted in a range of early caregiving risk within the sample (for additional description, see [42]). These longitudinal mother–child dyads (58.9% male child) were seen twice during infancy (infant mean ages = 12.62 months, SD = 0.65, and 18.55 months, SD = 1.02), again when they were 7–8 years old (M age = 7.62 years, SD = 0.32), and again at 19–20 years old (M age = 19.62 years, SD = 1.38).

Procedures were approved by the Hospital Institutional Review Board. Written informed consent for all assessments was obtained from the parent and (in late adolescence) from the adolescent.

### 2.2. Measures

**Infancy: Sociodemographics.** Early demographic risk was indexed by a score (0–5) summing the presence of the following 5 demographic risk factors in infancy: (1) no maternal high school diploma, (2) government aid recipient, (3) no partner in the home, (4) mother under age 20 at birth of first child, and (5) more than 2 children under the age of 6 in the home. Minority race/ethnicity of child and parents were also provided by the mother.

**Infancy: Home observation of maternal interaction.** Naturalistic mother–infant interaction was observed in the family’s home at infant ages 12 and 18 months. Mothers were asked to engage in their typical routines and interact with the infant as they usually would. A 40 min segment of mother–infant behavior was videotaped and later coded in 10 4 min intervals using the Home Observation of Maternal Interaction Rating Scale (HOMIRS; [43]). The HOMIRS consists of 12 5-point rating scales, including Sensitivity, Warmth, Verbal Communication, Quality and Quantity of Comforting Touch, Quality and Quantity of Caretaking Touch, Interfering Manipulation, Covert Hostility, Anger, Disengagement, Flat Affect, and Time Out of Room. For a detailed description of each scale, see [42]. All scales were reliably coded, r_i_s = 0.76–0.99, and coders were naïve to all other data.

Lyons-Ruth and colleagues conducted a principal component analysis (PCA) on the above scales, which yielded two similar major factors at 12 and 18 months [42]. Factor 1 was labeled maternal involvement and Factor 2 was labeled hostile intrusiveness. For more details regarding the PCA, see [42]. These two factor scores, maternal involvement and hostile intrusiveness, are used in the present analyses.

**Infancy: Infant attachment disorganization.** At infant age 18 months, mothers and infants were videotaped in the Strange Situation Procedure (SSP; [19]). The SSP consists of eight structured 3 min episodes in which the mother leaves and rejoins the infant twice. The SSP is designed to be mildly stressful in order to activate the infant’s attachment behavioral system. SSP videos were coded for the three organized attachment classifications (secure, avoidant, and ambivalent) as well as disorganized attachment. Videotapes were reliably coded for secure and insecure classifications both by a computerized coding program and a trained coder. Agreement was reached on 86% of the tapes. As previously reported [44], agreement on the disorganized-disoriented classification between a senior coder and a second coder for 32 randomly selected tapes was 83% (*r* = 0.73). Coder reliability for the 9-point Level of Disorganized Behavior Scale was *r* = 0.84. Following earlier precedent, overall security of attachment was indexed by a three-level ordinal variable (1 = secure; 2 = insecure-organized; 3 = insecure-disorganized). The attachment distribution in infancy was secure 30%; insecure-organized 18%; disorganized 52%.

**Infancy: Maternal disrupted communication.** At infant age 18 months, maternal disrupted interaction with her infant during the SSP [19] was scored by two trained coders using the Atypical Maternal Behavior Instrument for Assessment and Classification (AMBIANCE) coding system [44]. The AMBIANCE codes for five dimensions of disrupted interaction: (1) *Affective communication errors*, defined as contradictory affective signals to the infant or inappropriate responses to the infant’s cues; (2) *role confusion*, defined as the mother soliciting the infant’s attention or affection in ways that override the infant’s signals; (3) *frightened-disoriented behavior*, defined as fearful, hesitant, or deferential behavior toward the infant or as disoriented behavior; (4) *negative-intrusive behavior*, defined as harsh or critical verbal communication and/or physical behavior; (5) *withdrawing behavior*, defined by creating physical or emotional distance from the infant. Frequencies of behaviors in each of the five AMBIANCE dimensions were coded. In addition to the frequency codes for the five dimensions, the coder assigned a 1–7 rating of the overall level of maternal disrupted interaction. Ratings of 5 or above are considered to indicate disrupted maternal interaction. Reliability between two coders (*N* = 15) ranged from r_i_ = 0.73–0.84 for the frequency scores for the five dimensions of the AMBIANCE, *K* = 0.93 for the rating of overall level of disrupted communication, and *K* = 0.73 (87% agreement) for the classification as disrupted/not disrupted [44]. Both the five dimension frequency scores and the continuous overall rating were used in the current analyses. Validity of the AMBIANCE in relation to infant disorganization and stability of the AMBIANCE over periods up to 5 years have been established by meta-analysis [27].

**Middle Childhood: Middle Childhood Disorganization and Control**. At 7 years of age, mother and child behavior was observed in the laboratory during a standard attachment assessment, characterized by a 5 min reunion following a 1 h separation [28]. Child attachment behavior was coded using the middle childhood disorganization and control scales (MCDC; [45]). The MCDC rates the extent of child controlling-punitive, controlling-caregiving, and disorganized behavior on three separate 9-point scales. The controlling-punitive scale assesses the extent to which the child displays hostile behavior toward the parent, marked by a challenging, humiliating, cruel or defying quality. The controlling-caregiving scale codes child caregiving behavior characterized by structuring, guiding, entertaining, and organizing the interaction with the parent. The disorganized scale codes odd, out-of-context behavior, such as lack of a consistent interaction strategy, unpredictable, confused behavior, and indicators of disorientation or dissociation. See [29] for more detailed description of the scales. All codes were reliable, r_i_ = 0.83–0.97, *N* = 22, and coders were naïve to all other data.

**Middle Childhood: Emotional Availability.** Maternal behavior during the 5 min mother–child reunion was coded using the emotional availability (EA) coding system [46]. Maternal EA was scored from the EA Scales Scoring Manual (2nd edition), which has three maternal scales. Maternal sensitivity ratings (1–9) were based on positive affect sharing, awareness, and timely responsiveness to child behavior. Hostility scores (1–5) included both overt (e.g., name calling and threats) and covert (e.g., eye rolling and sighing) behavior. The maternal non-intrusiveness scale (1–7) was curvilinear, with lower scores representing intrusive behavior and higher scores representing passive-withdrawn behavior (i.e., optimal scores were at the midpoint), which made interpretation of mean scores difficult. Inspection of the data indicated that mothers in this sample were coded predominately in the passive-withdrawn range (51.16% elevated toward passive withdrawal versus 13.95% elevated toward intrusiveness). Another 34.80% of participants were coded at the scale midpoint as neither intrusive nor passive withdrawn. For clear interpretation of results, the scale was converted to a unidirectional scale for passive withdrawal by combining the 13.95% coded in the intrusive direction with the 34.80% coded at the scale midpoint to represent the lowest scale point indicating ’no passive withdrawal’. Higher scores then indicated elevations in passive withdrawal only. See [47] for more details regarding the EA coding. All codes were reliable (r_i_ = 0.95–0.98), and coders were naïve to all other data.

**Late Adolescence: The Goal-Corrected Partnership in Adolescence Coding System.** At age 19, adolescents and their mothers were videotaped during a 5 min unstructured reunion and 10 min discussion of a conflict in their relationship. The Goal-Corrected Partnership in Adolescence Coding System (GPACS; [30]) was used to code the security of the interaction between the participant and his or her parent. The GPACS coding system includes ratings on 10 5-point scales. The collaborative communication scale rates the extent to which the interaction is cooperative, reciprocal, and balanced for the dyad as a whole. The other nine scales rate the behavior of the adolescent or the parent separately, including four scales that rate forms of adolescent disorganized or controlling behavior, four scales that rate corresponding aspects of parental behavior, and a final scale for parental validating behavior. All codes were reliable (r_i_ = 0.75–0.96, *N* = 16) and coders were naïve to all other data.

The adolescent caregiving-role-confused behavior scale assesses the extent to which the adolescent attempts to manage or take care of the parent or modulate the parent’s behavior. The adolescent hostile-punitive behavior scale assesses the extent to which the adolescent attempts to control the parent through hostile, punitive, or devaluing behavior toward the parent. The adolescent odd, out-of-context behavior scale codes the extent to which the adolescent engages in odd, out-of-context, or contradictory behaviors, which may seem disjointed, startling, or inexplicable. Lastly, the distracted-disoriented scale measures the extent to which the adolescent exhibits distracted, disoriented, or inwardly absorbed behavior.

The scale for parent’s validation rates the degree to which the parent supports the adolescent’s exploration of thoughts and feelings related to the conflict. The parental role-confusion scale assesses the extent to which the parent fails to assume a parental role by failing to structure the interaction, failing to contribute to the task goals (i.e., discussing the conflict), or remaining excessively self- focused. The scales for parental hostile-punitive behavior, odd, out-of-context behavior and distracted-disoriented behavior are parallel to the adolescent scale described above. See [48] for a more detailed description of the GPACS coding system.

In the initial validation study [30], confirmatory factor analysis was applied to the 10 scales of the GPACS, resulting in a four-factor model that included dyadic factors for (1) collaborative interaction, (2) hostile-punitive interaction, (3) odd-disoriented interaction, and (4) caregiving-role-confused interaction. The current analyses first explored the four dyadic factor scores, followed by examination of the separate adolescent and parent subscales.

**Infancy to Age 18: Severity of childhood maltreatment.** As recommended by recent studies [49,50], overall severity of CM from infancy to age 18 was coded from multi-method indices of maltreatment that included documented CPS involvement, young-adult reported abuse, and coder-rated extent of maltreatment experiences derived from Adult Attachment Interviews (AAIs) administered in late adolescence. The overall severity of maltreatment rating (7-point scale) was generated by reviewing the following measures collected over the course of a 20 year study: (1) state protective services involvement or placement in foster care between 0 and 18 years; (2) adolescent reports of abuse on the Conflict Tactics Scale-2nd version (CTS-2; [51]); (3) adolescent reports of abuse on the Traumatic Stress Schedule (TSS; [52]); and (4) coder ratings of the extent of maltreatment revealed on the Adult Attachment Interview, coded by the Childhood Traumatic Experiences Scales-Revised (CTES-R; [53]).

CPS involvement of families in this study was primarily for neglect of the study child or, in the case of infants, for abuse or neglect of an older child. The CTS-2 is a widely used 78-item measure of tactics used during conflict between family members, including physically and emotionally abusive behavior. The TSS is an eight-question narrative survey probing traumatic experiences. Responses to three TSS questions on experiences of sexual or physical assault were reviewed in rating overall severity of CM. The CTES-R consists of four 5-point scales for rating the severity of physical abuse, sexual abuse, verbal abuse, and witnessed interpersonal violence coded from the AAI. The AAI is a semi-structured, transcribed, one-hour interview that asks participants in detail about attachment-related experiences with primary caregivers. The administration of the AAI in this study included additional questions about experiences of sexual or physical abuse, using the Antecedent Experiences Questionnaire [54]. Reliabilities between two coders on the four scales of the CTES-R were r_i_ = 0.89–0.98 (*N* = 56). Any discrepancies were resolved by discussion.

Based on review of the above materials, each individual’s overall severity of maltreatment from birth to age 18 was classified into one of seven levels: 1 = no occurrence of maltreatment; 2 = harsh punishment only; 3 = witnessed domestic violence only; 4 = verbal abuse only; 5 = physical abuse (using state guidelines for abuse), sexual abuse (using state guidelines for abuse), or protective services/foster care involvement; 6 = two under Level 5; 7 = all those under Level 5. The occurrence of physical and sexual abuse was judged according to state Department of Social Services guidelines for maltreatment. For example, for physical abuse, repetitive experiences of being hit other than on the buttocks or the hand by a primary caregiver were rated 5. Any sexual contact with a child younger than 16 was considered sexual abuse. Harsh punishment alone or witnessed violence alone did not lead to placement in Level 5. If maltreatment meeting state guidelines was noted as present on any of the assessments, it was considered present even if not reported on other types of assessment. Coding of the overall severity of maltreatment scale was highly reliable as assessed by intraclass correlation between two raters on 37 randomly selected protocols, r_i_ = 0.99. Any discrepancies were resolved by discussion. Previous work has shown that this severity scale has good validity in relation to various forms of young adult psychopathology [33,55,56].

It should be noted that we did not have consistent data on perpetrator or chronicity of maltreatment. Therefore, our measure of severity was defined by the dual criteria of reaching state guidelines for abusive treatment and by polyvictimization (number of types of abuse experienced). The nature of this definition of severity should be kept in mind when interpreting results.

### 2.3. Statistical Analyses 

Descriptive and preliminary correlational analyses to explore potential covariates were conducted using IBM SPSS Statistics 25 (IBM, Armonk, NY, USA). Multiple linear regression analyses were then conducted to assess the independent associations between severity of CM and mother and child behavior in infancy, childhood, and late adolescence. These individual regression analyses were followed up with models that included all significant variables from a given period (i.e., infancy, middle childhood, and late adolescence) in a single model, in order to assess overlapping variance across measurement subscales in relation to severity of CM. Regression analyses were conducted in Mplus Version 8 [57] using full information maximum likelihood (FIML) and bootstrapping to account for missing data, non-normality, and small sample size [58]. Bias-corrected confidence intervals (CIs) were used to assess significance; 95% CIs that do not contain zero are significant at *p* < 0.05.

## 3. Results

### 3.1. Descriptive and Preliminary Analyses 

Descriptive information for continuous study variables is included in Table 1. Categorically, 35.5% of the sample did not experience any type of harsh or abusive treatment, 11.8% of the sample reported harsh punishment only, 10.0% reported emotional-verbal abuse only, 26.4% experienced one type of abuse (physical abuse, sexual abuse, protective services/foster care involvement), 15.5% experienced two types of abuse, and 0.9% experienced three types of abuse. None of the participants witnessed domestic violence only, without also experiencing some additional form of abuse. Child sex, ethnicity, and demographic risk in infancy were explored as potential covariates in relation to overall severity of CM. There were no significant correlations (r_s_ = −0.019 to 0.069, *p* > 0.62). Thus, demographic factors were not controlled for in regression analyses predicting severity of CM.

### 3.2. Prediction of Severity of CM from Maternal and Child Behavior during Infancy 

Separate linear regression models, with bootstrapped CIs, were conducted to assess maternal and child behavior during infancy as risk indicators of severity of childhood maltreatment by age 18 (Table 2). First, overall infant attachment security to disorganization (coded secure = 1/insecure = 2/disorganized = 3) was not significantly related to the severity of childhood maltreatment by age 18 (Table 2). In addition, the scaled score for extent of disorganization did not reach significance in relation to severity of CM by age 18 using bootstrapped confidence intervals (Table 2). However, it is important to note that the association between disorganization and CM was moderate in size and that the *p* value, without bootstrap CIs, was significant (*β* = 0.271, *p* = 0.051).

Regarding *maternal* behavioral risk indicators, maternal hostile intrusiveness during home observations at both 12 and 18 months were significantly related to the extent of CM by age 18. In contrast, the degree of the mother’s involved interaction with the infant at home at 12 or 18 months was not related to later severity of CM.

In the lab observation at 18 months, the mother’s classification as disrupted in her communication with her infant in the attachment assessment was also a significant indicator of greater severity of CM by age 18 (Table 2). Given this association, the five AMBIANCE frequencies for aspects of disrupted interaction were also examined. Only maternal withdrawal at 18 months was significantly related to greater severity of CM by age 18. Notably, maternal negative-intrusiveness in the SSP assessment was not significant, suggesting that the attachment assessment in the lab, which was more likely to activate infant distress, elicited a different set of maltreatment-related maternal behaviors than did the naturalistic home observation (Table 2).

While these separate analyses give a detailed picture of which maternal and infant behaviors are related to CM, it is also important to know which behaviors overlap and which behaviors are unique risk indicators. Therefore, a second analysis was conducted by simultaneously entering all variables previously significantly related to CM. With all significant risk indicators in a simultaneous model, only maternal withdrawal was a unique indicator of CM (*β* = 0.305, *SE* = 0.130, CI = 0.012, 0.532). Other variables were no longer significant, including extent of maternal hostile intrusiveness at home at 12 months (*β* = 0.233, *SE* = 0.192, CI = −0.563, 0.189) and 18 months (*β* = 0.261, *SE* = 0.142, CI = −0.027, 0.532). These results indicate that maternal withdrawal from the infant’s attachment cues is a particularly important indicator, accounting for unique variance in association with CM. However, it is also important to note that in this simultaneous model, maternal hostile intrusiveness at home yielded effect sizes that might reach significance in a larger sample, suggesting the potential importance of this indicator as well.

### 3.3. Prediction of Severity of CM from Maternal and Child Behavior in Middle Childhood

Separate linear regression models were conducted to assess maternal and child behavior during middle childhood as risk indicators of severity of CM by age 18 (Table 3). When the three aspects of disorganized/controlling child attachment behavior at age 7 were assessed, attachment disorganization at age 7 was significantly associated with greater severity of CM by age 18, while extent of caregiving-controlling behavior and punitive-controlling behavior were not significantly associated with CM (Table 3).

Regarding maternal emotional availability in middle childhood, only maternal passive-withdrawal at age 7 was associated with severity of CM (Table 3). When both the maternal and child variables from middle childhood contributing significant variance were simultaneously entered into a single model, only maternal passive-withdrawal accounted for unique variance in severity of CM (*β* = 0.322, *SE* = 0.142, CI = 0.023, 0.573). Although child disorganization at age 7 was no longer a significant risk indicator with bootstrapped CIs (*β* = 0.334, *SE* = 0.139, CI = −0.005, 0.555), child disorganization yielded a medium effect size which was significant in the non-bootstrapped results (*β* = 0.334, *p* = 0.016), suggesting that child disorganization might make an unique contribution in a larger sample.

### 3.4. Associations Between Severity of CM and Maternal and Child Behavior in Late Adolescence 

Linear regression models were also conducted to assess the associations between quality of parent–adolescent interaction and severity of CM by age 18 (Table 4). Fifty-four additional participants were added to the study at age 19. Therefore, results are presented first for the longitudinal participants only (*N* = 56), for direct comparison with the results in infancy and middle childhood, and then for the full adolescent sample of *N* = 110, to take advantage of the additional power in the larger sample. 

The four GPACS dyadic factor scores were examined first, since these were the most inclusive variables, encompassing both parent and adolescent contributions to interaction (Table 4). In the smaller longitudinal sample, collaborative communication in the dyad was significantly negatively associated with severity of CM, while odd-disoriented interaction in the dyad was positively associated with severity of CM (Table 4). In this smaller sample, caregiving-role-confused dyadic interaction and hostile-punitive interaction did not reach significance in relation to severity of CM (Table 4).

When these analyses were repeated with the greater power afforded by the larger sample, lack of collaboration in the dyad and greater odd-disoriented behavior in the dyad were again significant (Table 4). In addition, in this larger sample, hostile-punitive interaction in the dyad was also significantly associated with severity of CM.

Given significant associations with the dyadic factor scores, the parent and adolescent ratings contributing to the factor scores were examined individually for their relations to CM. Results for both the longitudinal cohort and the larger sample are shown in Table 4. For the full sample (*N* = 110) in late adolescence, adolescent hostile-punitive behavior, caregiving behavior, and odd, out-of-context behavior were all significantly associated with greater severity of CM (Table 4). Adolescent distracted-disoriented behavior was the only scale not associated with severity of CM. For the smaller longitudinal sample, neither the adolescent’s caregiving behavior nor distracted-disoriented behavior reached significance (Table 4).

Regarding the parent scales for the full sample (*N* = 110), parental validation of the adolescent was significantly negatively related to the severity of CM, while the parent scale for odd, out-of-context behavior was positively associated with severity of maltreatment. The parent’s hostile-punitive behavior, role-confused behavior and distracted-disoriented behavior in interaction with the adolescent did not reach significance (Table 4). For the smaller longitudinal sample, the parent’s odd, out-of-context behavior did not reach significance, while the closely-related scale of parent distracted-disoriented behavior did reach significance (Table 4).

Finally, when a simultaneous model was conducted on the three significant dyadic factors (*N* = 110) to assess unique contributions, only the odd-disoriented factor was uniquely associated with severity of maltreatment (*β* = 0.237, *SE* = 0.106, CI = 0.014, 0.437). This indicates that collaborative and hostile aspects of interaction occurred primarily in concert with odd-disoriented interaction, rather than having separate and unique relations to severity of maltreatment. Simultaneous analysis of all significant adolescent and parent individual scales (*N* = 110) yielded a similar result. When all significant scales were entered into a simultaneous model, only the adolescent’s odd, out-of-context behavior (*β* = 0.243, *SE* = 0.085, CI = 0.073, 0.409) made a significant unique contribution to the association with severity of maltreatment. No other aspects of adolescent or parent behavior made additional contributions (parent: odd, out of context (*β* = 0.040, *SE* = 0.098, CI = −0.182, 0.211); validation (*β* = −0.065, *SE* = 0.115, CI = −0.295, 0.155); adolescent: hostile-punitive (*β* = 0.169, *SE* = 0.108, CI = −0.052, 0.370); caregiving (*β* = 0.064, *SE* = 0.101, CI = −0.113, 0.279)).

## 4. Discussion

The aim of the present study was to identify features of parent–child interaction throughout development that might serve as indicators of elevated risk for CM. Mother–child interaction was observed across multiple assessments during infancy, middle childhood, and late adolescence. Results point to specific aspects of parent–child interaction in each of these developmental periods that are associated with increased severity of CM by age 18.

### 4.1. Findings by Developmental Period

Among the assessments in infancy, infant attachment disorganization was only marginally associated with overall CM by age 18, in that the significant result did not survive bootstrapping. Prior meta-analyses have confirmed that infant disorganized attachment is associated with concurrent CM [20,21] but few studies have examined infant disorganization as a potential risk indicator of overall CM throughout childhood and adolescence. This study indicates that, given the robust effect size, infant attachment disorganization shows promise as an important risk indicator of overall severity of childhood maltreatment by age 18, but future work in larger samples will be needed to confirm its role.

In infancy, assessments of maternal behavior were the best risk indicators of overall severity of CM by age 18. Maternal hostility in the home at both 12 and 18 months, as well as maternal withdrawal in the lab observation, were associated with the severity of CM by age 18. Interestingly, neither lack of involvement at home nor negative-intrusiveness in the lab were significantly correlated with CM. Notably, Wilson et al. demonstrated meta-analytically that maltreating parents were more likely to show aversive (e.g., hostile-intrusive) behavior when observed for a longer period of time in the home environment, compared to shorter laboratory assessments [40]. Given that the home observation was 40 mi, compared to 25 min for the lab observation, this may contribute to our results.

The mildly stressful attachment assessment in the lab highlighted the importance of maternal withdrawal as a risk indicator of severity of CM, and, notably, only maternal withdrawal in the SSP accounted for unique variance in severity of CM, when other significant indicators were controlled. Both infant distress and infant attachment behavior were more highly activated by the mildly stressful lab procedure, so that the lab assessment may have highlighted maternal responses to the child’s attachment behavior that were less salient at home. These differential findings from home and lab underscore the importance of observing parenting in both settings when assessing infant risk.

By middle childhood, both maternal and child behaviors were significant risk indicators of severity of CM by age 18. Among the child assessments, only child attachment disorganization at age 7 was significantly related to severity of CM. In contrast, child punitive-controlling and caregiving-controlling attachment behaviors were not. The behaviors that characterize attachment disorganization in middle childhood are analogous to those characterizing attachment disorganization in infancy, that is, odd, out-of-context, or anomalous behaviors during the interaction with the parent, such as sudden frantic skipping around the room, slipping into baby/nonsense talk, freezing or stilling, etc. Other work beyond infancy in both high-risk [59] and low-risk samples [60] also indicates that, compared to child controlling behavior, disorganized attachment behavior is associated with more severe family risk factors. This importance of disorganization at age 7 as a risk indicator of overall maltreatment by age 18 is an important and novel finding. Prior studies of attachment and CM have focused on associations with current documented maltreatment and have primarily included samples younger than four years of age (meta-analyses: [20,21]). Thus, the current finding extends this literature by showing that attachment disorganization assessed in middle childhood is also an important indicator of the severity of CM by age 18. The significant effect of middle childhood disorganization underscores the importance of assessing the child’s attachment behavior beyond infancy, in that the child’s continued inability to organize a consistent stance toward the attachment figure into middle childhood becomes a robust indicator of CM.

In middle childhood, maternal behavior was also a significant risk indicator of severity of CM by age 18. Similar to the lab results in infancy, in middle childhood, maternal passive-withdrawal, but not maternal hostility or insensitivity, was related to severity of CM. Because home observations were not conducted in middle childhood, we do not know whether hostility at home might also have been a significant indicator, as in infancy. Thus, from infancy through age 7, maternal passive withdrawal emerges as a salient risk indicator of severity of CM by age 18.

Also, importantly, in middle childhood as in infancy, when both maternal and child indicators were included in the simultaneous model, only the mother’s passive withdrawal accounted for unique variation in CM. This greater importance of early maternal behavior compared to early child behavior as an indicator of long-term outcomes has also been found in other long-term longitudinal studies [61,62,63].

Theoretically, maternal withdrawal might be important because it signals the inability of a more passive parent to protect the child from abuse by others. Maternal withdrawal might also indicate more emotional disconnection from the child, thereby elevating risk for maternal emotional abuse, neglect, and other forms of maltreatment. Alternately, withdrawal may be a marker for the parent’s own history of CM, signaling a fear that any emotional closeness to the child might trigger anger and abusive behavior. Further work is needed to evaluate these potential mechanisms of effect. Notably, early maternal withdrawal has also emerged as a unique predictor in developmental trajectories toward adolescent borderline personality disorder features, suicidality/self-injury, and substance abuse [61,64,65]. Notably, all of these forms of psychopathology have been further associated with CM [66]. These converging findings underscore the importance of assessing parental withdrawal in future work.

Finally, it is unclear why parental hostility in middle childhood was not shown to be an indicator of severity of CM by age 18, given that a large number of studies have linked parental hostility to concurrent documented maltreatment in childhood [40,67]. It is possible that neglect was a more salient aspect of maltreatment in this sample than in some others, or that the attachment-focused assessment in middle childhood, which included a one-hour separation before the 5 min reunion episode, elicited more child arousal and display of disorganized behavior that worked against the mother’s display of harsh or hostile behavior. Finally, the lab-based nature of the assessment may have pulled for maternal withdrawal more than for maternal harsh or hostile behavior, which might be more likely displayed in an unstructured home observation, as noted by [40].

By late adolescence, in the larger sample that included both longitudinal and cross-sectional participants, three aspects of dyadic interaction were associated with severity of CM, namely, less collaboration, more hostile-punitive behavior, and more disorientation. This suggests that by late adolescence, global interaction difficulties are seen among dyads in which the adolescent has been exposed to more severe maltreatment. Notably, in contrast to middle childhood, hostile-punitive interaction was again differentially associated with maltreatment, as it was in infancy. However, in infancy, it was the parent’s hostility that was salient, while in late adolescence, it was the *adolescent’s* hostility that was contributing most strongly to the association with severity of maltreatment. Notably, CM was measured through age 18 only, so these observations at age 19 occurred after the period in which childhood maltreatment was assessed. Thus, they may be best viewed as sequalae linked to earlier maltreatment.

Strikingly, in late adolescence when all dyadic factors were entered into a single regression model, odd-disoriented behavior in the dyad was the primary factor accounting for unique variance in severity of CM. In addition, when the *individual* parent and adolescent scales were entered into a single regression model, only the adolescent’s odd, out-of-context behavior emerged as a unique correlate. This indicates that, among the parent and adolescent behaviors assessed, adolescent odd, out-of-context behavior carried the most information about the severity of maltreatment experienced by age 18. One implication of this unique status is that the assessment of odd, out-of-context behavior in adolescence should be prioritized when risk of maltreatment is the focus of the work. To date there is little work with high-risk samples that has coded these odd, out-of-context behaviors in adolescence. For one exception, Khoury et al. coded young adults with borderline personality disorder (BPD), depression only, and no diagnosis and found that odd, out-of-context behaviors were more marked among young adults with BPD [68], a diagnostic group repeatedly shown to have elevated rates of CM [66,69].

### 4.2. Developmental Trends: Continuity and Discontinuity

The developmental progressions seen across time in these data also deserve comment. First, we did not find consistent trajectories of *child* hostility toward the parent across development. Infant anger was rare and was not assessed. At age 7, child punitive-controlling behavior was assessed but was not associated with severity of CM. However, by late adolescence, more hostile-punitive adolescent interaction was associated with greater severity of CM. It may be that, earlier in development, the child is afraid of the parent and inhibits direct hostility toward the parent. However, numerous longitudinal studies, including the present one, have documented prediction from the parent’s hostile interaction in early childhood to the child’s conduct problems and hostile interactions with peers by school-age [70,71,72]. Thus, children exposed to early parental hostility in infancy may inhibit hostility towards parents in early development, while expressing increased hostility towards peers and teachers at school. Finally, by late adolescence the maltreated adolescent has become a more active partner in hostile transactional processes with the parent. Given that causal influence cannot be inferred from our correlational results, an important future direction will be to use Actor Partner Interdependence Models [73] and other modeling techniques to assess direction of influence. These analyses would give more insight into how much the parent or child or both are influencing the other partner’s behaviors in interactions from infancy to late adolescence.

Second, there was developmental continuity across infancy and childhood in the salience of maternal passive-withdrawal as a risk indicator of more severe maltreatment by age 18. The centrality of caregiver withdrawal as an indicator of severity of CM is an important aspect of these findings, because parental lack of involvement is less often assessed in studies of maltreatment compared to hostile, harsh, or aversive parental behavior [40]. However, the meta-analysis by Wilson et al. highlighted that, among currently maltreating parents, parental maltreatment is also associated with parental lack of involvement, particularly during unstructured tasks and among neglectful parents [40]. Of further interest, maternal withdrawal in infancy and childhood have been found to be associated with caregiving-controlling behavior on the part of the child by middle childhood [29,65]. This longitudinal association is not surprising, given that abdication of a parental role is central to both parental withdrawal and child caregiving-controlling behavior (i.e., parent–child role reversal). This abdication, in turn, appears to draw some children and adolescents into attempting to maintain the parent’s involvement by taking on a caregiving role, including structuring, praising, and entertaining the parent [29]. However, child caregiving behavior at age 7 was not related to severity of CM, so adopting a caregiving stance toward a withdrawing parent may not occur as readily in the context of maltreatment or may not take shape until late adolescence among some maltreated youth. Future work is needed to better understand the familial and psychological mechanisms that link parental withdrawal and child caregiving behavior in developmental trajectories among maltreated youth.

A final notable aspect of the developmental trends across time was the consistency of the relation between indices of attachment disorganization and severity of maltreatment by age 18. Markers of attachment disorganization, in the form of odd, out-of-context behaviors in the presence of the parent, were associated with increased severity of CM—first, marginally in infancy, and then strongly in middle childhood and late adolescence. In fact, by late adolescence, the adolescent’s odd, out-of-context behavior was most strongly associated with severity of CM, after all other significant aspects of parent–adolescent interaction were controlled. Thus, odd, out-of-context behavior deserves inclusion in future developmental work assessing the context and correlates of child maltreatment.

### 4.3. Limitations

Important limitations of this work should also be noted. First, given the relatively small sample size, the results should be replicated in larger samples to assess generalizability. Second, in late adolescence, the assessment of observed parent and adolescent behavior at age 19 occurs subsequent to the assessed period of childhood maltreatment up to age 18. Therefore, additional studies are needed to assess the same aspects of interaction earlier in adolescence in relation to severity of maltreatment. This is quite feasible because the GPACS assessment has been validated from age 14 onward [32]. Third, this study did not assess the child’s interaction with the mother’s male or female partners due to the high rate of single parents in the study. Nor were we able to assess the perpetrator of the experienced maltreatment. Thus, we are unable to assess the extent to which fathers or other caregivers contributed to these findings, either as perpetrators of abuse or as protective figures who provided more secure relationships [74,75,76]. Future work is needed to assess how interaction with other caregivers across development might be associated with reduced or elevated risk for CM. Also importantly, our measure of severity of CM relied on two criteria for severity: meeting state guidelines for abuse and polyvictimization. Other ways of defining severity might yield somewhat different results. In particular, given the multi-method nature of this assessment, it was not possible to specify with any certainty the specific time periods during which maltreatment occurred. Future research is needed to assess CM across different stages of development, in order to ascertain whether particular maternal and child behaviors, at different time points, are differentially related to the occurrence of CM at various stages of development. However, this study also has several strengths, including the use of direct observational assessments, a multi-faceted index of CM severity, and a prospective longitudinal design spanning from infancy to late adolescence.

## 5. Conclusions

In summary, multiple aspects of parent–child interaction in each developmental period were associated with the severity of CM by age 18. We also found consistency over time in regard to two important indicators, maternal withdrawal and child disorganization. In both infancy and middle childhood, maternal withdrawal made unique contributions to the estimate of severity of CM. In addition, child disorganized behavior showed consistent value across middle childhood and late adolescence as a risk indicator or correlate of severity of CM. Neither of these aspects of behavior, as yet, have been as routinely assessed in studies of CM, in comparison to more often studied forms of aversive parenting. Identification of a set of observable risk indicators that can be reliably assessed across infancy, childhood, and late adolescence offers the possibility of improving early identification and provision of preventive supports for families at elevated risk for CM, regardless of whether maltreatment itself is documented by social service workers.

## Figures and Tables

**Table 1 ijerph-17-03749-t001:** Descriptive characteristics for continuous measures.

	M (SD)	Range
**Infancy:**		
**Infant attachment security (18 months)**		
Overall security of attachment ^a^	n/a	1–3
Disorganized attachment behavior ^b^	6.00 (2.51)	1–9

**Maternal disrupted communication (18 months) ^c^**		
Overall disruption (continuous)	4.24 (1.73)	1–7
Affective communication errors	5.24 (4.44)	0–20
Role confusion	4.91 (6.85)	0–27
Disorientation	3.00 (3.32)	0–16
Negative-intrusive behavior	2.27 (3.33)	0–14
Withdrawal	3.18 (3.34)	0–15

**Maternal behavior at home (12 months)**		
Involvement factor	0.59 (0.94)	−2.18–1.75
Hostile intrusiveness factor	0.0027 (1.01)	−2.79–1.35

**Maternal behavior at home (18 Months)**		
Involvement factor	0.0014 (0.94)	−2.09–2.15
Hostile intrusiveness factor	−0.045 (0.92)	−1.38–2.90

**Middle childhood (age 7):**		
**Child disorganized-controlling attachment**		
Caregiving-controlling	2.68 (2.16)	1–9
Punitive-controlling	2.77 (1.84)	1–8
Overall disorganization	1.94 (1.97)	1–9

**Maternal emotional availability**		
Mother sensitivity	3.29 (2.18)	0–7
Mother hostility	0.47 (0.75)	0–2
Mother passive withdrawal	1.18 (1.00)	0–3

**Late adolescence (age 19):**		
**Dyadic GPACS ^d^ factors**		
Collaboration factor	2.70 (0.86)	2–5
Hostile-punitive factor	3.02 (1.08)	1.5–6
Disoriented factor	4.48 (1.74)	3.25–10
Caregiving-role-confused factor	3.22 (1.47)	1.5–6.5

**Adolescent GPACS scales**		
Hostile-punitive interaction	2.13 (0.88)	1–4
Odd, out-of-context behavior	1.45 (0.83)	1–5
Distracted-disoriented interaction	1.43 (0.72)	1–4
Caregiving-role-confused interaction	2.02 (1.11)	1–5

**Parent GPACS scales**		
Validation of adolescent	2.66 (0.84)	1–4
Hostile-punitive interaction	1.96 (0.81)	1–4
Odd, out-of-context behavior	1.40 (0.74)	1–4
Distracted-disoriented interaction	1.28 (0.58)	1–3
Role-confused interaction	2.21 (1.04)	1–4

**Severity of childhood maltreatment (birth to age 18)**	3.45 (2.16)	1–7

Note. *N* = 56 (data refer only to the longitudinal sample). ^a^ Security of attachment coded as secure = 1; insecure-organized = 2; insecure-disorganized = 3. ^b^ Extent of disorganization rated 1–9. ^c^ The AMBIANCE subscales are based on frequency counts, whereas the AMBIANCE overall scale is based on a rating. ^d^ The Goal-Corrected Partnership in Adolescence Coding System.

**Table 2 ijerph-17-03749-t002:** Regression results for associations between mother–child interactions during infancy and severity of childhood maltreatment by age 18.

	Severity of Childhood Maltreatment			
	β	Std. Error	Beta (Unstd)	Bootstrap CI [95%]
**Infant attachment behavior (18 months)**				
Overall security of attachment ^a^	0.045	0.126	0.000	−0.205, 0.275
Disorganized attachment behavior ^b^	0.271	0.139	0.466	−0.018, 0.535

**Maternal behavior at home (12 months)**				
Involvement	0.023	0.141	0.053	−0.244, 0.298
Hostile intrusiveness	0.428 *	0.137	0.920	0.654, 0.112

**Maternal behavior at home (18 months)**				
Involvement	−0.218	0.132	−0.498	−0.450, 0.072
Hostile intrusiveness	0.419 **	0.098	0.983	0.193, 0.584

**Maternal disrupted communication in the SSP (18 months)**				
Disrupted vs. non-disrupted	0.316 *	0.140	1.354	0.015, 0.577
Affective communication errors	0.279	0.143	0.136	−0.039, 0.528
Role confusion	−0.053	0.137	−0.017	−0.328, 0.199
Disorientation	0.055	0.158	0.036	−0.289, 0.328
Negative intrusive	0.028	0.143	0.018	−0.242, 0.321
Withdrawal	0.352	0.118	0.228	0.085, 0.553

Note. *N* = 56. Results are based on separate regression analyses. ^a^ Security of attachment coded as secure = 1; insecure-organized = 2; insecure-disorganized = 3. ^b^ Extent of disorganization rated 1–9. * *p* < 0.05 based on 95% CI; ** *p* < 0.01 based on 99% CI.

**Table 3 ijerph-17-03749-t003:** Regression results for associations between mother–child interactions during middle childhood and severity of childhood maltreatment by age 18.

	Severity of Childhood Maltreatment			
	β	Std. Error	Beta (Unstd)	Bootstrap CI [95%]
**Middle childhood disorganization and control (MCDC) scales**				
Caregiving-controlling	0.057	0.177	0.058	−0.320, 0.367
Punitive-controlling	−0.031	0.178	−0.037	−0.400, 0.285
Disorganization	0.402 *	0.133	0.445	0.075, 0.612

**Maternal emotional availability scales (EAS)**				
Maternal lack of sensitivity	0.315	0.170	0.314	−0.051, 0.606
Maternal passive-withdrawal	0.392 **	0.137	0.857	0.094, 0.633
Maternal hostility	0.066	0.183	0.193	−0.319, 0.403

Note. *N* = 56. Results are based on separate regression analyses. * *p* < 0.05 based on 95% CI; ** *p* < 0.01 based on 99% CI.

**Table 4 ijerph-17-03749-t004:** Regression results for concurrent associations between indices of parent–adolescent interaction and severity of childhood maltreatment.

	Severity of Childhood Maltreatment			
	β	Std. Error	Beta (Unstd)	Bootstrap CI [95%]
**Longitudinal Sample (*N* = 56)** **Dyadic GPACS Factors**				
Collaboration factor	−0.308 *	0.141	−0.775	−0.555, −0.018
Hostile-punitive factor	0.287	0.135	0.572	−0.008, 0.525
Odd-disoriented factor	0.369 **	0.116	0.456	0.114, 0.567
Caregiving-role-confused factor	0.195	0.145	0.286	−0.123, 0.454

**Parent GPACS scales**				
Validation of adolescent	−0.322 *	0.142	−0.827	−0.579, −0.026
Hostile-punitive interaction	0.119	0.150	0.318	−0.193, 0.402
Odd, out-of-context behavior	0.228	0.129	0.663	−0.071, 0.437
Distracted-disoriented interaction	0.326 *	0.119	1.217	0.036, 0.517
Caregiving-role-reversed interaction	0.171	0.145	0.356	−0.131, 0.448

**Adolescent GPACS scales**				
Hostile-punitive interaction	0.492 **	0.112	1.212	0.235, 0.679
Odd, out-of-context behavior	0.311 *	0.118	0.810	0.032, 0.502
Distracted-disoriented interaction	0.170	0.155	0.512	−0.155, 0.443
Caregiving-role-confused interaction	0.197	0.160	0.382	−0.152, 0.480

**Cross-sectional sample (*N* = 110)**				
**Dyadic GPACS factors**				
Collaboration factor	−0.269 **	0.088	−0.583	−0.433, −0.093
Hostile-punitive factor	0.252 **	0.088	0.414	0.075, 0.420
Disoriented factor	0.311 **	0.098	0.397	0.089, 0.476
Caregiving-role-confused factor	0.184	0.097	0.244	−0.014, 0.362

**Parent GPACS scales**				
Validation of adolescent	−0.266 **	0.093	−0.588	−0.447, −0.081
Hostile-punitive interaction	0.176	0.094	0.399	−0.007, 0.356
Odd, out-of-context behavior	0.206 *	0.087	0.621	0.011, 0.357
Distracted-disoriented interaction	0.127	0.111	0.452	−0.097, 0.336
Caregiving-role-confused interaction	0.147	0.098	0.273	−0.052, 0.327

**Adolescent GPACS scales**				
Hostile-punitive interaction	0.305 **	0.084	0.618	0.128, 0.462
Odd, out-of-context behavior	0.338 **	0.076	0.865	0.179, 0.474
Distracted-disoriented interaction	0.093	0.104	0.287	−0.122, 0.281
Caregiving-role-confused interaction	0.220 *	0.101	0.406	0.023, 0.416

Note: Results are based on separate regression analyses. * *p* < 0.05 based on 95% CI; ** *p* < 0.01 based on 99% CI.

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
