# Peer review of "Aspects of Parent–Child Interaction from Infancy to Late Adolescence are Associated with Severity of Childhood Maltreatment through Age 18"

_ijerph, 2020, doi:10.3390/ijerph17113749_

Round 1

Reviewer 1 Report

This paper is of great interest both for the subject, methodology (longitudinal), data based on observation (not questionnaires), and the results obtained. It is also very well structured and very complete, especially in the rationale for the work, presented in the introduction, as in the description of the measures (section 2.2). It is also worth highlighting the presentation of results and discussion, with a very rigorous Limitations section (4.2).

However, I consider that it could improve in some very specific aspects.

Line 48-49. Where it says: (US DHHS, 2020; Finkelhor, Turner, Ormond, & Hamby, 2013); It should say: (Finkelhor, Turner, Ormond, & Hamby, 2013; US DHHS, 2020).

Line 120. It says: "history of CPS involvement", but the initials CPS have not been entered before in the text. I suppose it refers to "Child Protective Services", but for an international understanding of the text it should be made explicit.

Line 351, Table 1, first column. The "Late Adolescence (n = 56)" section clearly indicates the number of dyads. However, it is not indicated in the sections "Infancy" and "Middle Childhood". Although it can be understood that it is a clarification since for "Late Adolescence" there are two samples or cohorts -longitudinal and cross-sectional-, it is confusing that the number of dyads in all cases is not specified. It could also be clarified by indicating in the title of the Table that the data refer only to the longitudinal cohort (n = 56). 

Line 358-359-360. It says: "However, the scaled score for extent of disorganization was significantly related to severity of childhood maltreatment reported by age 18 (β = .271, SE = .139, CI = .030, .486; Table 2)". Although Table 2 shows exactly the same data for the Disorganized attachment behavior variable, the table does not indicate that the results are significant or the level of significance (they are not marked with asteristics).

Line 569. Where it says: (eg, Widom et al., 2009; Reich et al., 1997); It should say: (e.g., Reich et al., 1997; Widom et al., 2009).

Line 582. Where it says: ...Wang & Kenny, 2014; Stormshak, Bierman, McMahon, & Lengua, 2000); It should say: ...Stormshak, Bierman, McMahon, & Lengua, 2000; Wang & Kenny, 2014).

Two additional suggestions are included, relating to two sections of the paper.

Section 2.1, Participants. Although the information is complete and pertinent, reading this section is somewhat confusing. At the time of the first reading, the inclusion of a second sample or cohort (cross-sectional) does not seem relevant. Until the Results section is read, specifically section 3.4, the strategy is not well understood. Could more precise information on study design be included in section 2.1, or add a specific section on design?

Section 2.2, Measures. This section is very complete, the measures are carefully described, with sufficient information on the situations and observation procedures, as well as the coding procedures. The use of measurements based on observation, and not on questionnaires, is undoubtedly one of the main contributions of this paper. However, it is very extensive. I do not know to what extent it would be possible to reduce this section without reducing the quality of the information it provides. It's just a suggestion.

Reviewer 2 Report

attached a letter

Reviewer 3 Report

Thank you for the opportunity to review the manuscript, “Aspects of Parent-Child Interaction from Infancy to Adolescence Predicting Severity of Childhood Maltreatment through Age 18.” The task of identifying observable risk factors for child maltreatment is an incredibly valuable line of study. The findings highlight the need for greater attention to subtle aspects of parent-child interactions that are associated with risk for maltreatment and may otherwise be missed. It is notable that neither maternal hostility nor punitive control predicted CM severity. Despite the great theoretical importance of this work, there are numerous methodological concerns that should be addressed including the conceptualization of the CM measure as “severity,” consideration of adolescent-parent interactions as a predictor, and the question of directionality between parent-child variables and CM severity. Detailed information about these and other concerns are provided below.

  • The literature review is very thorough, however, adolescent-parent relationships and adolescent “odd, out of context behavior” could use more support/background in the introduction. I am aware research in this area is limited, but wherever possible, more support should be added to support the examination of this construct.

  • The participants section is a bit hard to follow given there are two separate samples. I think part of the confusion is that it is framed around the endpoint of the study where the “children” are young adults when in the longitudinal study they began as infants. Thus, in the first sentence of the methods, some clarification is needed to make it clear to which person in the dyad (mother or child) the term “young adulthood” pertains to – are you studying adaptation and psychopathology in the mothers or the children? Is the mean age of 19 years for the mothers or the “children” – I first assumed the mothers since 19 is not really a child, but then realized if 59% are female it must be the children. Since the sentence mentions a dyad, I suggest providing details about both parties, clearly marked. An alternative suggestion, remove the stats in the first sentence and leave the numbers for when you discuss each sample separately.

  • Greater justification is needed for reframing the non-intrusiveness scale as passive-withdrawn. If higher scores could mean either intrusive or passive/withdrawn, more info is needed about how many participants scored on the passive side of the curve. If it was all of them, this should be stated. If not, what percent scored on the intrusive side and how were their data managed in analyses? Please expand on statement: “new scores were calculated.”

  • The multi-method assessed of CM severity is a strength of the study. However, details are needed on how combined retrospective and prospective reports were combined/reconciled. Were there discrepancies between sources of information and if so, how were they managed?

  • Somewhat related to point 4, it is arguable whether this measure captures severity as it does not account for the type of experience within each category (e.g., some types of physical abuse may be more “severe” than others, some types of sexual abuse may be more severe than some types of physical abuse and vice versa) or the frequency of the event. Rather, this is a measure of CM occurrence and polyvictimization. This should be considered when discussing findings.

  • Is the range provided for the AMBIANCE dimensions the range of frequency counts? I would have expected them all to range from 1-7. Are the M and SD also for the frequency count? Whichever measure was used needs to be more explicit either in the table or in the measure description, or both. Currently, the methods states that the subdimension ratings were used.

  • Please provide frequency and percent for all 7 levels of the CM severity measure.

  • Were any maternal demographic factors explored as covariates? If not, it may be helpful to explain why.

  • Lines 390-391 – Please clarify what the two Beta’s refer to – are there before and after controlling for other variables? If so, perhaps state .272 vs. .217. If not, please clarify what relationship each Beta pertains to. Upon further reading, they appear to be for 12 and 18 months? If so, I do not think they need to be repeated here given they are stated a few lines up.

  • In tables 2, 3, and 4 it may be helpful to clearly indicate in the title or note that the results are for the separate regressions and not the full model.

  • On page 13, be consistent in use of n versus N.

  • The term predictor is leading to difficulty in the interpretation of results. For example, infant disorganized attachment predicted severity of CM but I am guessing the authors do not intend to suggest that disorganized attachment leads to greater severity of CM. Rather, it is a risk indicator. This issue comes up in the use of adolescent variables to predict CM severity in later point. Perhaps a difference in wording is needed to more accurately portray results and help readers understand how these various aspects of the parent-child relationship across time are associated with higher severity of CM. It’s the implying of a temporal relationship that is difficult to tease apart. Middle childhood attachment disorganization is not necessarily a predictor in that its occurrence leads to later CM severity. How can one tease apart whether greater CM severity predicts disorganization or greater disorganization predicts CM severity?

  • If you have information on when during development the abuse events occurred, that may be helpful to present to help readers gauge the degree to which middle childhood and adolescence variables are truly predictors or more so correlates (see next point). It would be interesting to dig deeper into predictors of type and timing of abuse since you appear to have the data to answer these sorts of questions. Are maternal or parent-child relationship factors across different periods of time stronger predictors of different types of abuse or abuse at particular times? For example, does withdrawal in infancy predict abuse only during early childhood or across development? Does it predict the accumulation of occurrences of abuse? I understand these questions may be outside the scope of the present study but may be important future directions of this work.

  • Relatedly, given that abuse is more common in early childhood, particularly compared to adolescence, and that the adolescent measure came at age 19, it is important to consider whether the adolescent-parent relationship is truly a predictor of severity by age 18 or rather an artifact of earlier abuse. This is briefly noted as a limitation but this relationship should not be assessed in this backwards direction. Adolescent variables could be considered correlates but not predictors. Much more caution should be taken in the discussion in pointing out that adolescent-parent relationships may be more or a correlate than a predictor. It is arguable whether this is also true for the middle childhood variables.

  • Lines 519-536 should be combined into a single paragraph

  • Given that the predictors for all three developmental time points were not analyzed in a single regression model (rather they were examined separately within each developmental stage), the conclusion (line 640-641) that these variables across development were “uniquely predictive” is not quite accurate.

  • Consider consistently using the term “late adolescence” throughout (including in the title) given the participants were 19. The terms adolescence and late adolescence are both used throughout.

Round 2

Reviewer 3 Report

The authors should be commended for their quick, thoughtful, and thorough response to reviewer feedback. I appreciate the care and concern that were taken for each point raised and do not have any further suggestions for improvement. I wish the authors all the best in their future work on this important topic.